# Infant Acute Myeloid Leukemia: A Unique Clinical and Biological Entity

**DOI:** 10.3390/cancers13040777

**Published:** 2021-02-13

**Authors:** Charlotte Calvo, Odile Fenneteau, Guy Leverger, Arnaud Petit, André Baruchel, Françoise Méchinaud

**Affiliations:** 1Pediatric Hematology and Immunology Department, Robert Debré University Hospital (APHP and Université de Paris), 75019 Paris, France; charlotte.calvo@aphp.fr (C.C.); andre.baruchel@aphp.fr (A.B.); 2Hematopathology Department, Robert Debré University Hospital (APHP and Université de Paris), 75019 Paris, France; odile.fenneteau@aphp.fr; 3Pediatric Immunology, Hematology and Oncology Department, Trousseau Hospital (APHP and Université Paris Sorbonne), 75012 Paris, France; guy.leverger@aphp.fr (G.L.); arnaud.petit@aphp.fr (A.P.)

**Keywords:** acute myeloid leukemia, infant, childhood, AML

## Abstract

**Simple Summary:**

Infant acute myeloid leukemia (AML) is a rare subgroup of AML of pediatric patients under two years of age. For a long time, they have been included in pediatric forms of AML. However, infant AML demonstrates unusual clinical and biological characteristics, and its prognosis differs from AML in older children. Most treatment protocols do not distinguish age subgroups in pediatric AML, when in reality these unusual forms of infancy might benefit from specific targeted therapeutics. Herein we summarize the various specificities of infant AML.

**Abstract:**

Infant acute myeloid leukemia (AML) is a rare subgroup of AML of children <2 years of age. It is as frequent as infant acute lymphoblastic leukemia (ALL) but not clearly distinguished by study groups. However, infant AML demonstrates peculiar clinical and biological characteristics, and its prognosis differs from AML in older children. Acute megakaryoblastic leukemia (AMKL) is very frequent in this age group and has raised growing interest. Thus, AMKL is a dominant topic in this review. Recent genomic sequencing has contributed to our understanding of infant AML. These data demonstrated striking features of infant AML: fusion genes are able to induce AML transformation without additional cooperation, and unlike AML in older age groups there is a paucity of associated mutations. Mice modeling of these fusions showed the essential role of ontogeny in the infant leukemia phenotype compared to older children and adults. Understanding leukemogenesis may help in developing new targeted treatments to improve outcomes that are often very poor in this age group. A specific diagnostic and therapeutic approach for this age group should be investigated.

## 1. Introduction

Infant acute myeloid leukemia (AML) is a rare disease, historically defined by AML occurring under 1 year of age. This definition has been extended to AML in children up to 2 years of age as patients aged from 1 to 2 years old demonstrate similar clinical and biological pattern of disease [1]. 

In this review we will focus on de novo infant AML, even though infant AML associated with Down syndrome represents a significant component of this subgroup. We will describe all striking features of infant AML. Rare and atypical clinical presentations in babies have been described for a long time. Similarly, pathologists have reported distinctive distribution of FAB subgroups with predominance of acute megakaryoblastic leukemia (AMKL) and acute monocytic leukemia [1]. AMKL is highly associated with this subgroup and is rare in other age groups. This intriguing feature has prompted interest by different groups and recent in-depth molecular studies have been reported in the literature [2,3,4]. These research groups have been able to decipher the genomic landscape of AML and more specifically AMKL, by identifying recurrent fusions either unique to infant AML or strongly associated with this age group. They have also shown the paucity of mutations compared to adult AML [3,4]. Further, they have interrogated the molecular specificities of this subgroup, modeling different genetic fusions in mice [5,6]. They have been able to show the role of ontogeny and fetal hematopoiesis development in the emergence of infant AML. During fetal life hematopoiesis, the fetal liver supports self-renewal and differentiation of hematopoietic stem cells and multipotent progenitors (HSC/MPPs). Oncogenic events, mainly fusions, occurring in these fetal liver progenitors impact leukemia phenotype [5,7]. How these fusions impact transcription, disrupt transcription factor balance and promote leukemogenesis may reveal opportunities for treatment. Thus, targeted approaches could interfere with oncogenic determinants to improve treatment efficiency of infant leukemia often characterized by very poor outcome [5,7].

## 2. Epidemiology

### 2.1. Age

Infant acute myeloid leukemia (AML) has been conventionally defined as AML in patients aged <1 year [1,2,8]. If infant specific features can be seen up to 3 years, they are more prominent up to 2 years of age and many groups agree to define the infant group by children less than 2 years of age [1,9]. The characteristics of infant AML specificities may be related to developmental processes and fetal hematopoiesis features that are still expressed beyond the age of one.

### 2.2. Incidence 

AML is a very rare disease but is the leading cause of childhood leukemic mortality [3,10,11]. Infant AMLs are as frequent as infant ALLs, even though AML is 5-times less frequent than ALL in older children [12]. The incidence of pediatric AML is estimated to be between 5 and 7 cases per million people per year, with a peak of incidence at 11 cases per million people per year at 2 years of age [10]. Infant AML is a distinct entity, representing 10–25% of pediatric AML, depending on age limits: 32% of children if the cut off is set at 3 years old in a Japanese cohort, 12% of children if <1 or 24% <2 years old in a ELAM02 French cohort [1,2]. Overall AML in children less than 2 years of age represent a fourth of childhood AML with unique features.

A large number of inherited conditions predispose children to the development of AML but explain only a small percentage of all AML cases. Down syndrome is by far the leading predisposing syndrome, but others are described [10]. We will not discuss AML associated with genetic syndrome in this paper.

## 3. Clinical Presentation 

Typical leukemia presentations with anemia, bleeding, febrile neutropenia, bone pain and organomegaly are not uncommon but some specific aspects need to be underlined. AML in infants is more frequently associated with higher white blood cell counts and extra-medullary symptoms than in older children, notably skin lesions and central nervous system (CNS) involvement [1,2,8,12,13]. 

A lot remains unknown regarding the causality of these specific clinical features in infant leukemias, but development and fetal hematopoiesis and its different origins might play a role. 

Neonatal acute myeloid leukemia has striking features. Congenital non-Down syndrome leukemia are very rare, accounting for less than 1% of pediatric leukemia and occurring within the first 4 weeks of life [12,14,15]. They often present with clinical symptoms: hepatomegaly, splenomegaly and skin lesions (leukemia cutis). Jaundice, ascites and pleural effusions are also fairly common while lymphadenopathy is less so. Hepatic failure can lead to death despite regression of the leukemia [14]. Central nervous system (CNS) infiltration is reported in around 50% of all cases and manifests with a bulging fontanelle, papilledema and retinal hemorrhages as well as a reduced level of consciousness. Cutaneous infiltrates are particularly a feature of neonatal AML, being present in about two-thirds of such babies and can occur without any peripheral blood or bone marrow involvement [14]. Typically, leukemia cutis manifests as generalized firm nodules, which may be blue, red, brown or purple. The appearance has been described as a ‘blueberry muffin rash’ [12,14,15]. Prognosis is very often dismal. Spontaneous remissions have been observed, notably in AML associated with t(8;16)(p11;p13) [15,16]. These spontaneous remissions are specific of this period associated with a significant shift in hematopoiesis, from liver to marrow, reinforcing importance of stage of development in AML occurrence [14,16]. 

Leukemia cutis (LC), also known as cutaneous myeloid sarcoma is not limited to neonatal period and remains common in infant leukemia. LC is commonly seen in FAB-4 and 5 AML [17,18,19]. Patients present with diffuse and papulonodular erythematous or violaceous skin lesions corresponding to leukemia infiltration into the epidermis, dermis or subcutaneous tissues [20].

In the French ELAM02 cohort, skin involvement was significatively more prevalent in infants occurring in 14.5% of children less than 2 years old compared to 2.6% of older children [1,19]. This presentation is associated with other extra-medullary organ involvement in AML and is negatively associated with prognosis [17,18,19]. One hypothesis is that chemotherapy sufficient to induce remission in bone marrow may not be sufficient to penetrate the skin, thus leading to a greater incidence of relapse in those patients.

Liver involvement may be prominent in infants and sometimes dominates clinical presentation and raises diagnosis discussion with other liver solid tumors like hepatoblastoma [21,22]. Liver failure may aggravate clinical presentation and limit chemotherapy [21,22].

CNS involvement is also more common: in the ELAM02 cohort, 26% of patients <2 years of age presented with CNS positivity, in contrast with 12% of patients >2 years old [1]. In the Children’s Oncology Group COG experience, infants less than 2 years of age represented 35% to 45% of patients with CNS involvement defined as CNS3 [23,24]. Several hypotheses have been proposed to explain CNS involvement in infants: (i) the immaturity of the blood-brain barrier, which does not fully develop until the age of six months, (ii) the greater vasculature of the leptomeninges of infants and preschool children [1], and (iii) the high frequency of monoblastic leukemias, as these cells are able to penetrate directly through the cytoplasm of the endothelial cells into the brain [1,13].

Intracerebral chloroma is not uncommon in this population and CNS imaging should be a standard in infant leukemia. This clinical presentation is associated with a higher risk of relapse but does not seem to affect overall survival [1,23]. 

Overall, extramedullary involvement is common and very diverse anatomical presentations are described, without bone marrow involvement in some cases. These manifestations could be linked to the embryology of hematopoiesis. In fact, infant AML may arise during fetal life, occurring while hematopoiesis is not yet medullary, but located in skin and then liver [25]. Solid tumors can be the at the forefront of the diagnostic discussion and anatomical pathology evaluation needs to be broad and include hematopoietic markers.

## 4. Cytologic and Cytogenetic Characteristics

### 4.1. Morphology

If all French–American–British (FAB) classification subtypes can be seen in infants, it is striking how acute monocytic (M5) and megakaryoblastic (M7, AMKL) leukemias dominate in the first year, while decreasing with age. Conversely, there is a low proportion of AML without and with maturation, M1 and M2 subtypes, in infants compared to older children for whom they represent the most common subtype [1,2].

Acute monocytic leukemia represent a significant group of infant AMLs and have classical morphological features and immunophenotypic characteristics.

On the other hand, AMKL can be really difficult to diagnose. It is not rare to have very few leukemic cells in peripheral blood. Blasts can also be very rare in bone marrow, and difficult to identify in abundant immature physiological population. Moreover, bone marrow aspiration can be difficult because of myelofibrosis with limited material for evaluation [26,27]. AMKL blasts show cytoplasmic blebs, cell clumping and binucleation. Blasts can be quite specific but rare and clustered resembling solid tumor metastasis (Figure 1).

AMKL blasts are immunophenotypically positive for CD33, CD36, CD41, CD42b, and CD61 [26,27]. In some AMKL group overexpression of CD56 (NCAM1) and under-expression of both HLA-DR and CD38 is found and described as RAM phenotypes associated to specific genetic subtypes [28,29,30].

AML in infants does not always fulfill cytologic criteria of >20% of blast infiltration in bone marrow. AML can be diagnosed in these patients if low number of leukemic blasts in bone marrow is associated with the presence of recurrent cytogenetic abnormalities [31]. 

### 4.2. Cytogenetics 

In the infant group, rearrangement of *KMT2A* (formerly *MLL*) is the most common cytogenetic alteration, representing more than 50% of the cases while core binding factor (CBF) rearrangements are rare (<5%) [1] (Table 1). 

Rare translocations specific for this age group including t(1;22) and t(7;12) can be identified by cytogenetic. Fluorescence in situ hybridization (FISH) will identify *KMT2A* rearrangement but does not always resolve the identity of the fusion partner. The presence of an additional copy of chromosome 21 raises the hypothesis of mosaicism and signals the need to check the constitutional karyotype. 

However, most rearrangements are cryptic and are not identifiable by conventional cytogenetic. A normal karyotype is common and additional techniques including targeted RT-PCR, RNA-sequencing, and potentially whole genome sequencing are required to fully characterize infant AML [32].

Overall, there is no clear correlation between morphology, cytogenetics and molecular lesions.

## 5. Genomics 

High throughput sequencing has helped decipher infant leukemia and revealed striking differences with older children and adult leukemia. The Children’s Oncology Group (COG)–National Cancer Institute (NCI) TARGET AML initiative was able to characterize the genomic landscape of almost 1,000 pediatric AML patients by whole genome sequencing of samples from 197 patients and targeted sequencing of tumor cells from 800 patients [3]. Others have also explored more limited group such as AMKL [4,33]. AMKL is better understood but fusions found in this subtype are also found in other phenotypes and are not unique. Genomic alterations appear to be more strongly associated with age rather than morphologic. 

### 5.1. Rearrangements 

Infants usually exhibit unfavorable molecular genetic factors compared to older children. Conversely, favorable CBF rearrangements are rare in this group of patients with RUNX1-RUNX1T1 (t(8;21)(q22;q22)), or CBFB-MY11 (inv16(p13;q22)) and PML-RARA (t(15;17)(q24;q21)) translocations reported rarely [1,13].

#### 5.1.1. KMT2A Rearrangements 

Wild-type *KMT2A* is involved in transcriptional regulation and chromatin modifications for the establishment of cell-specific transcriptional programs, with a major role in embryogenesis and maintenance of embryonic and adult hematopoiesis. When disrupted due to a translocation, the crucial *KMT2A* regulatory domains (e.g., DNA binding, histone marking/recognition, transactivation) is fused to a partner gene. Different partners are encountered, of which *AF4/FMR2* family member 1 (*AFF1*, a.k.a. *AF4*), *MLLT3* super elongation complex subunit (*MLLT3*, a.k.a. *AF9*), *MLLT1* super elongation complex subunit (*MLLT1*, a.k.a. *ENL*), and *MLLT10* histone lysine methyltransferase DOT1L cofactor (*MLLT10*, a.k.a. *AF10*) are the most prevalent [34]. Most partners are also regulators of transcription by direct or indirect interaction with RNA polymerase II. The resulting KMT2A chimeras are capable of subverting crucial transcriptional machinery, altering global gene expression and epigenetic signatures of the affected cells. This ultimately results in strongly enhanced and improper expression of genes involved in proliferation and lineage identity, conferring stem cell-like properties and consequent transformation [35]. Among impacted genes the *HOX* gene cluster plays a crucial role.

Infant acute leukemia is frequently associated with translocations involving the *KMT2A* gene at 11q23, with approximately 40–60% of infants with AML harboring *KMT2A* rearrangements (e.g., MLL-ENL, MLL-AF4, or MLL-AF9) [1,2,8,13,35,36]. Most of these KMT2A -R AML are morphologically classified as FAB-M4 or M5 [37]. Still, *KMT2A* fusion can be seen with AMKL and there is no absolute correlation between phenotype and genotype.

*KMT2A* has been shown to rearrange with more than 80 distinct partner genes, which do not have the same implication for patients [37]. In fact, outcome of patients significantly differs depending on the partner of *KMT2A*, unlike KMT2A-R ALL. Patients with a t(1;11)(q21;q23) showed excellent outcome independent of other risk factors (5-year EFS 92% +/− 5%), whereas those with t(10;11)(p12;q23), or t(10;11)(p11.2;q23) showed poor clinical outcome (5-year EFS 29% +/− 13% and 32% +/− 5% respectively) independent of other factors [32,37,38]. The incidence and prevalence of individual subtypes vary with age; thus some KMT2A partners are more likely to be found in older patients (e.g., t(6;11)(q27;q23) or t(11;17)(q23;q21)), while for its other partners there is no differences in age at diagnosis [35,37].

The translocation t(9;11) is the most frequent translocation identified in infants and is associated with intermediate risk [13,38]. 

As the 11q23/MLL-rearranged group has an impact on prognosis, FISH screening for KMT2A rearrangements at diagnosis has become the standard approach in many AML protocols. Some partners are not identified by this method; therefore, additional techniques such as RT-PCR and soon RNA sequencing should become part of standardized screening procedures to correctly identify patients with specific low- or high-risk KMT2A rearrangements [37].

#### 5.1.2. CBFA2T3-GLIS2 

*CBFA2T3* was initially identified as a fusion partner with *RUNX1* in therapy-related AML and shown to facilitate transcriptional repression, as well as being implicated in hematopoietic stem cell quiescence [29]. On the other hand, *GLIS2* is closely related to the GLI subfamily and likely functions in modulating the hedgehog signaling pathway [29]. The fusion of *GLIS2* to *CBFA2T3* as a result of inv(16)(p13.3q24.3) leads to increased expression of the GLIS2 DNA-binding domain [39]. Although GLIS2 is not normally expressed in the hematopoietic system, its fusion to *CBFA2T3* as a result of the inv(16)(p13.3q24.3) results in high level expression of the C-terminal portion of the protein including its DNA-binding domain [39].

Cryptic inversion of chromosome 16 leading to the *CBFA2T3-GLIS2* fusion gene is the most frequently identified chimeric oncogene to date in AMKL patients, even though it is not restricted to this FAB sub-type [28,29,40]. Due to the cryptic nature of the *CBFA2T3-GLIS2* fusion, this lesion is not identifiable by morphology or cytogenetics, but a large proportion of patients can be identified by flow cytometry with the combination of CD56/HLA- DR/CD38, CD56/HLA-DR and CD56/CD38, called RAM phenotype [28,29,30,41]. The fusion protein directly regulates overexpression of CD56, by binding to its promoter.

The CBFA2T3-GLIS2 blocks the differentiation of megakaryocytic cells, inhibiting the expression of *GATA1*, and increases self-renewal capacity by inducing the overexpression of *ERG*, thus priming the leukemic phenotype and blocking differentiation [28,29]. The impact of this fusion on the GATA1/ERG balance underline the crucial role of this equilibrium in normal hematopoiesis development [42].

When not associated with normal karyotype, *CBFA2T3-GLIS2* can be associated with trisomy 21, complex karyotype and hyperdiploidy, suggesting that the fusion oncogene can be found in patients with a variety of cytogenetic aberrations [28].

AML with *CBFA2T3-GLIS2* fusion gene is found only in patients aged <3 years, and it is the second most frequent fusion gene in patients aged <1 [2]. Extramedullary involvement is more frequent in patients expressing the *CBFA2T3-GLIS2* chimeric gene (25%) compared to the frequency reported for pediatric AML in general [28,41]. Extramedullary involvement can be initially confused with a non-hematopoietic tumor, especially in the presence of cranial bone, ribs and lumbosacral column involvement. 

*CBFA2T3-GLIS2* AMLs are associated with a dismal outcome even with high-intensity therapies, including allogeneic hematopoietic stem cell transplantation (allo-HSCT). Five-year overall survival (OS) ranges between 15 and 30% [2,8,28,39]. 

#### 5.1.3. NUP98-KDM5A

*NUP98*, located on chromosome 11p15, is a part of the nuclear pore complex (NPC), a multi-protein structure responsible for the transport of various macromolecules (e.g., RNA) into and out of the nucleus, and functions as a transcriptional regulator while also helping mitotic progression [43]. 

*NUP98* fusions, involving the N-terminal portion of *NUP98* and the C-terminal portion of the fusion partner, are present most frequently in childhood AML, representing about 5% of patients [6,43]. Gene expression profiling demonstrated a typical pattern of expression in *NUP98* fusions AML with overexpression of *HOXA* (similar to KMT2A-R) and *HOXB* cluster genes, defining a unique transcriptional signature of NUP98-rearrangements [4,6,44]. NUP98-KDM5A almost always carry mutations in *RB1* with decreased expression of this gene [4].

Many of these NUP98 fusions are cryptic on conventional cytogenetic analysis, and thus only the recent development of next-generation sequencing (NGS) technologies has revealed their high prevalence in pediatric leukemias.

NUP98-KDM5A is a recurrent cryptic fusion (t(11;12)(p15;q35)) found in about 10% of infant AMKL and is associated with dismal prognosis [2,44]. 

#### 5.1.4. HOX Rearrangements

Fusions involving *HOX* cluster genes (HOX-R) were reported in a study examining 98 childhood AMKL using genomic sequencing [4]. *HOX* fusions with overexpression of *HOX* represent more than 10% of AMKL and fusions are either able to enhance self-renewal or result in loss of function of these regulatory transcripts [4]. This underlines the role of *HOX* deregulation in leukemogenesis in this age group.

#### 5.1.5. t(1;22)(p13;q13) RBM15–MKL1

Acute megakaryoblastic leukemia associated with t(1;22)(p13;q13) was first described in 1991 and results from the fusion of 2 transcription factors, *RBM15* and *MKL1* (also known as *OTT* and *MAL*) [45]. These 2 genes encode transcription factors. In fact, *RBM15* is involved in hematopoiesis and has been involved in the control of the activity of RBPJ (recombination signal binding for immunoglobulin KJ region), which plays an important role in the NOTCH pathway while *MLK1* is a cofactor of the SRF (serum response factor), implicated in the Rho GTPase/G-actine pathway [46]. In a mouse model, Mercher et al. demonstrated that RBM15-MKL1 induces abnormal differentiation of HSCs which differentiate toward megakaryoblastic lineage and develop AMKL [47].

This fusion has been found in 10 to 25% of AMKL [33], affecting infants at a particularly young age with a median around 4 months, and can be associated with hepatic fibrosis [12,14,27,48,49]. There is a higher prevalence of this translocation in females, and AMLs with t(1;22) are considered standard risk [2,27,49,50].

#### 5.1.6. t(7;12)(q36;p13) ETV6-HLXB9

*ETV6* has numerous translocation partners, can be rearranged in myeloid or lymphoid malignancies and can arise in utero leading to postnatal acute leukemia [36]. Translocations with *ETV6* normally result in the generation of a fusion gene where the helix-loop-helix domain of ETV6 plays an important role in activating or modulating the function or expression of the partner gene. In t(7;12)(q36;p13) *ETV6* is joined to the regulatory sequences and first exons of the *HLXB9* homeobox gene (currently called *MNX1*), which acts as a transcription factor. This fusion induces overexpression of *HLXB9*, interferes with normal hematopoietic differentiation and impacts cell maturation [36,51,52,53].

The t(7;12) is difficult to detect through conventional karyotyping because of the small sizes of the involved 7q and 12p. Therefore, diagnosis is facilitated by the use of FISH [36,51].

This rare translocation, often reported with trisomy 19 is associated with a very dismal prognosis [36,51,52,53].

#### 5.1.7. t(8;16)(p11;p13) 

This translocation results in fusion of *MYST3*, localized on 8p11, and CREB-binding protein (CREBBP), localized on 16p13. Both proteins have histone acetyltransferase activity and are involved in transcriptional regulation and cell cycle control [16,54]. This AML shows overexpression of *HOXA* genes but not *HOXB*, which is a pattern shared with *KMT2A* rearranged AML [54].

The t(8;16)(p11;p13) associated AML is a rare entity, occurring in >50% of the cases in infants and 28% of this subgroup AML occurs in newborns with a majority of female patients [11,16]. Patients have mainly myelomonocytic (M4) or monocytic (M5) AML. Erythrophagocytosis is also frequently observed [54]. Extramedullary disease is frequently associated with this translocation with mainly leukemia cutis, CNS involvement and granulocytic sarcoma involving the bone. Outcome is similar from other types of AML in children, and seemed to be intermediate, whereas in adults, dismal outcomes can be associated with higher frequencies of therapy derived AML [54]. Importantly this type of AML is associated with spontaneous remission in the congenital forms, suggesting that watch and wait strategy can be proposed in newborns. These patients should be monitored closely as there is a high risk of recurrence at an older age [11,16]. The mechanism of spontaneous remission of neonatal leukemia has not been investigated directly.

### 5.2. Mutations

Compared to other cancers, AMLs in children, like adult AMLs, are remarkable by their paucity of mutations. Where mutations do occur, they affect different genes from those observed in adult AMLs [2,3]. 

Patients with CBFA2T3-GLIS2, KMT2A-R or NUP98 fusions seem to have even fewer mutations than subjects without these fusions [3]. Among these mutations some are thought to be relevant in leukemogenesis. 

*GATA1* mutation is known to play an important role in Down Syndrome (DS) associated AMKL [10,12,32]. In a pediatric cohort of Non-Down Syndrome AMKL, mutation of *GATA1* was the most common recurrent mutation, with patients harboring truncating mutations in either exon 2 or 3 (11–13,5%) [4,33]. Of the ten patients described carrying *GATA1* mutation, none presented with clinical DS features and one patient was diagnosed with mosaicism. Because of the role of GATA1 in DS-AMKL, authors hypothesized the potential role of the DS critical region (DSCR) on chromosome 21; 9 of the 10 *GATA1* mutant had in fact amplification of DSCR, suggesting the same molecular cooperativity [4]. Furthermore, patients with *GATA1* mutations and no recurrent translocation seemed to have superior outcome, similarly to DS-AMKL patients [4]. Some described this group as DS like AMKL [42].

Different groups identified several cooperating mutations different from adult cohorts and identified significant association with some recurrent translocation: RAS pathway lesions with KMT2A rearrangements, *RB1* mutations leading to decreased expression in NUP98-KDM5A, *MPL* mutations and HOX rearrangements [4,38].

### 5.3. Gene Expression Profiling 

Thanks to gene expression profiling, distinct genomic profiles in Non-DS AMKL cohort cluster together: KMT2A-R, NUP98-KDM5A, HOX-R, CBFA2T3-GLIS2, GATA1 mutant (Figure 2) [4]. 

KMT2A-R and NUP98-KDM5Asubtypes showed upregulation of *HOX* cluster genes, demonstrating that about 50% of pediatric non-DS-AMKL patients carry a HOX gene expression program [4,6,50]. Cryptic CBFA2T3-GLIS2 inversion cluster away from other non-DS-AMKL [4]. Coenen et al. also demonstrated that AML with t(8;16) had a gene expression profile that clustered closed to KMT2A-R AML with selectively highly expressed *HOXA* but not *HOXB* genes [16].

These data demonstrated striking features of infant AML: (i) fusion genes are able to induce AML transformation without additional cooperation, (ii) there is a paucity of associated mutations and (iii) gene expression profile clusters in defined groups [4,5,29].

## 6. Modeling Infant Leukemia: Ontogeny and Infant AML

The discovery of specific molecular lesions in AML has shed light on both leukemogenesis and hematopoietic development. To understand the mechanistic role of these different molecular lesions, mouse models have been developed using gene editing to create transgenic mice. In parallel, patients derived samples have been used to generate xenotransplants in immunocompromised mice (PDX) [5,6,7]. These transgenic mice have helped understanding the role of the different fusions in leukemic transformation. They have also contributed to our understanding of the role of pre- and post-natal development in efficiency of transformation and phenotype.

Using a transgenic mouse model Okeyo-Owuor et al. explored transformation efficiency of the MLL-ENL (KMT2A-MLLT1) fusion [7]. KMT2A fusion proteins are potent oncoproteins that drive leukemogenesis in infants with very few cooperating mutations. This suggests that fetal or neonatal hematopoietic progenitors may be exquisitely sensitive to KMT2A fusion proteins. Their data showed that the efficiency of KMT2A-ENL driven AML initiation changed with age and peaked shortly after birth, thus demonstrating that developmental context plays an important role in shaping the genetic landscape of KMT2A driven leukemias [7].

Using an inducible transgenic mouse model, Lopez et al., developed a model that phenocopied the human AML CBFA2T3-GLIS2 disease: they demonstrated striking differences in latency, phenotype, and molecular wiring, depending on the ontogenic stage at which the driver fusion oncogene was activated [5]. Developmental stage in which the transcript arises determined the phenotype of leukemia: fetal liver derived hematopoietic stem cell (HSC) expressed megakaryoblastic marker CD41 while adult bone marrow derived HSC showed myeloid phenotype. As AMKL and AML can be seen with KMT2A rearrangement, they used in a similar way KMT2A-MLLT3 transgenic mouse model to show that ontogenic stage and HSC hierarchy affected the phenotype induced by this transcript [5]. They demonstrated that not only the stage of development is important to determine the leukemia phenotype but also the type of HSC in which the translocation would occur: expression of KMT2A-AF9 in long-term HSC led to AMKL while its expression in adult progenitors led to myeloid leukemia [5]. This is consistent with recent findings that normal fetal hematopoiesis is primarily controlled by multipotent HSCs, whereas adult hematopoiesis presents with a higher ratio of committed progenitors [5]. Aggressive pediatric AMKL seems to originate in fetal HSCs and, more globally, age- and phenotype-specific associations observed in human patients rely on changes in the cellular architecture during hematopoietic ontogeny. 

These studies have helped to understand the role of ontogeny in infant leukemogenesis and shed light on the specificities of infant leukemia. It also opens some insight on how to target specific genetic lesion.

## 7. Targeted Treatment 

Infant with AML are currently treated with the same protocols used for older children and adolescents. Some collaborative groups have defined AML<1 year old as a high-risk factor and thus intensify the treatment (e.g., Associazione Italiana Ematologia Oncologia Pediatrica, AIEOP LAM2013/01) and have adapted dosage for chemotherapy to take in account the unique pharmacokinetic and pharmacodynamic profiles due to the young age. However, no protocol has been designed for this specific subgroup even though we demonstrated here that they exhibit unique clinical and biological characteristics.

As shown above, some infant AMLs and more specifically some specific genetic subgroups have dismal outcome [55]. Furthermore, these patients are at increased risk of infection and chemotherapy related toxicities. Outcomes remain unsatisfactory for this category of patients, although interestingly, in the French cohort of patients treated with ELAM02 protocol, as well as in AIEOP and BFM (Berlin-Frankfurt-Münster) protocols, no difference in CR or induction failure rate or incidence of relapse was observed comparing infants to older children [1,31,56].

Therefore, knowledge of the molecular and genetic background is of most relevance in order to detect novel leukemia and patient-specific treatment targets [32]. The expression of fusion oncoproteins specifically in malignant cells represents an important opportunity for therapeutic targeting; this principle was first successfully demonstrated in the development of imatinib for individuals with chronic myeloid leukemia and other malignancies expressing the BCR-ABL1 fusion oncoprotein [43].

Cardin et al. were able to recapitulate phenotypic, genomic and transcriptomic characteristics of NUP98-KDM5A AMKL in xenografted mice [6]. They could therefore demonstrate that these AMKL reproduced in PDX mice upregulated the JAK-STAT pathway and could be effectively targeted by JAK inhibitors.

Krivtsov et al. demonstrated in vitro and in vivo efficacy of Menin inhibitor in KMT2A-driven AML [57]. This small molecule inhibits proliferation and induces cellular differentiation and apoptosis in KMT2A-R AML cell line and PDX models. The oral selective menin inhibitor KO-539 showed activity in preliminary findings of the ongoing first-in-human KOMET-001 trial in adult patients with relapsed/refractory AML [58].

Menin inhibitors could also be of interest in NUP98-R AML, knowing the role of chromatin remodeling in leukemogenesis driven by NUP98-R oncoproteins [43]. 

By identifying specific phenotype characteristics of CBFA2T3-GLIS2 AMKL, Smith et al. evaluated CD56 as a potential target for therapy [29]. They used an anti-CD56 antibody drug conjugate in patients’ samples and showed specific toxicity in vitro. 

These various studies demonstrate that understanding these rare forms of AML at a molecular and functional level could lead to targeted treatments that are applicable in infant AML resulting in greater treatment tolerance and efficacy. 

## 8. Conclusions 

Infants with AML represent a cohort of patients with specific clinical and biological features. They present with some remarkable differences in clinical presentation, cytology, cytogenetic and molecular lesions when compared to older children. Some of these features are unique to AML in children less than 2 years of age. To date, infant AMLs are not differentiated from childhood AMLs in terms of therapeutic approach as they are treated on the same protocols. Understanding their differences could enable the creation of new international protocols designed to better characterize these leukemias and offer new therapeutic approaches.

## Figures and Tables

**Figure 1 cancers-13-00777-f001:**
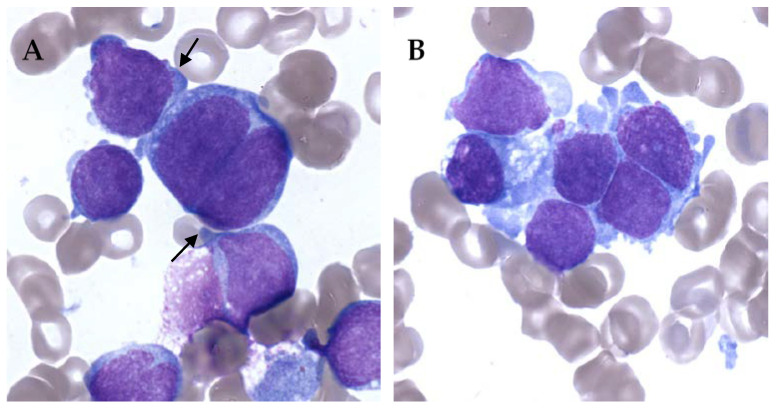
Bone marrow smears of acute megakaryoblastic leukemia (AMKL) patients, May–Grünwald–Giemsa staining, ×100. (**A**) Cytoplasmic blebs (black arrow) and binucleated cell. (**B**) Cell cluster giving a pseudo-solid tumor aspect of AMKL.

**Figure 2 cancers-13-00777-f002:**
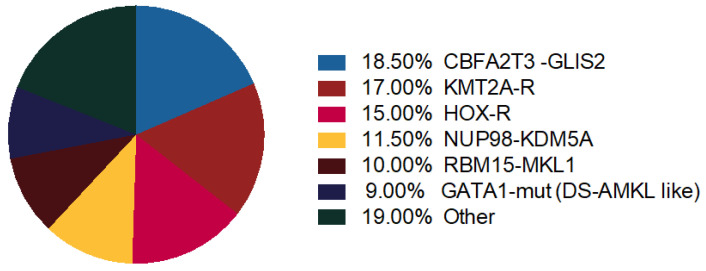
Molecular alterations in AMKL. Distribution of recurrent chromosome rearrangements and GATA1 mutations in pediatric non-Down syndrome (DS) AMKL [42,50].

**Table 1 cancers-13-00777-t001:** Distribution of cytogenetic abnormalities in infants <2 years old acute myeloid leukemia (AML) [1]. CBF, core binding factor.

Cytogenetic Abnormalities	%
*KMT2A* rearrangements	**55%**
Normal karyotype	9%
Adverse cytogenetic (monosomy 7, t(6;9)(p23;q34), complex karyotype)	14%
CBF rearrangements	5%
Others	17%

## Data Availability

Data sharing is not applicable to this review article.

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
