# Peer review of "Infant Acute Myeloid Leukemia: A Unique Clinical and Biological Entity"

_cancers, 2021, doi:10.3390/cancers13040777_

Round 1
Reviewer 1 Report
Calvo et al reviewd the current state of knowledge on infant AML in a well written review that provides a very comprehensiveoverview. Suggestions for targeted therapy were made and are appropriate - although at the very first stage of development.
Suggestions:
- Clearly this age group requires NGS - especially RNA-seq - as diagnostic approacg - I would stress that a little more than for example FISH
- Menin inhbitors may also be useful for NUP-rearranged leukemias, this can be added, although data are limited
- I suggest to add perhaps the French survival curves for infants versus oklder children to make the point whether their outcome is worse or not, and discuss utcome data for infant AML - as there are also series that state they do as well as older patients (remarkably)
Author Response
Please see in attachment

Reviewer 2 Report
The present manuscript by Calvo and colleagues provides an overview of the current knowledge on infant acute myeloid leukemia. The paper is overall well-written and detailed, I have some minor suggestions/comments.
1. The Authors consider as “infants” children below 2 years of age. Indeed, although historically the conventional definition of “infant AML” refers to AML diagnosed in a child before 1 year of age, over the years, many studies combined data of children <12 months of age with those of patient aged 12 to 24 months, due to similar features and survival outcomes. However, since for many collaborative groups the term “infant leukemia” refers to acute leukemia diagnosed in a child before 1 year of age, this should be clearly specified also in the abstract and/or title.
2. The Authors state that the treatment of infant AML is not clearly individualized by study groups. Actually, this is not completely true as an individualized approach is used by AIEOP for infant AML. Indeed age<1 year was introduced as a high risk criterion in the AIEOP LAM 2013/01 study protocol, leading to treatment intensification for this subgroup of patients.
3. I would suggest adding data on the outcome of infant patients in the various treatment protocols published so far (BFM, AIEOP…)
4. On page 3, it is stated that “Hepatic failure can lead to death despite regression of the leukemia” (line 96) and that “Cutaneous infiltrates are particularly a feature of neonatal AML, being present in about two-thirds of such babies” (line 99). I would suggest including bibliography to support these statements.
5. On page 3, lines 104-105, the sentence “These spontaneous remissions are specific of this period associated with a significant shift in hematopoiesis reinforcing the role of development in infant AML” is unclear. It would suggest rephrasing.
6. As the t(1;22)(p13;q13) translocation, leading to the OTT-MAL fusion protein, has been reported in association with better outcomes in infants with M7 AML, current published data on this subgroup of patients should be discussed.
7. On page 8, line 344, is the following sentence correct? “Importantly this type of AML is associated with spontaneous remission in the congenital forms, suggesting that watch and wait strategy can be proposed in newborns WITH A HIGH RISK OF RECURRENCE”.
8. On paragraph 7 (Targeted treatment, page 10, line 422) the Authors state that infant(s) with AML are currently treated with the same protocols used for older children and adolescents: see point 2.
9. The peculiarity of pharmacokinetic/pharmacodynamic profiles of certain chemotherapeutic agents (e.g., cytarabine) in infants, potentially increasing the risk of therapy-related toxicities and leading to the need of adapting dosages of cytotoxic drugs, should be discussed.
10. Lastly, I suggest editing the manuscript in order to revise minor spelling and grammar errors (e.g., on page 1 - Simple Summary:“infant AML demonstrateS peculiar clinical and biological characteristics, and ITS prognosis differs from AML in older children”.
Reviewer 3 Report
Please justify the use of the term "infant AML." is this really a distinct entity? As you wrote, infants with AML often have AMKL or monocytic leukemia, but I don't believe there is a category of AML called "infant AML."
Author Response
Please justify the use of the term "infant AML." is this really a distinct entity? As you wrote, infants with AML often have AMKL or monocytic leukemia, but I don't believe there is a category of AML called "infant AML."
As we explained in our review, infant acute myeloid leukemia (AML) has been historically defined as AML occurring under 1 year of age. This definition has been extended to AML in children up to 2 years of age as patients aged from 1 to 2 years old demonstrate similar clinical and biological pattern of disease. We are not referring to the FAB-classification when we use the term infant AML but to myeloid lineage.
As we explained extensively in our review these patients demonstrate unique clinical and biological features, that are different from AML occurring in older children. They also demonstrate peculiar pharcokinetic and -dynamic profiles to chemotherapy.
Reviewer 4 Report
With the rapid development of patient sample sequencing and expansion of molecular studies in animal models, emerging evidences of the specificity of infant acute myeloid leukemia have been shown recently, while there is a lack of summary of recent advances in the field. This Review, by Calvo and colleagues, is filling the knowledge gap. In the Review, authors summarized the various specificities of infant acute myeloid leukemia, including most recent progresses in the field, and suggested a specific diagnostic and therapeutic approach for this disease. The manuscript is well presented and the review does not have any concerns about it.
Round 2
Reviewer 3 Report
-